# Temporal Profiles and Dose-Responsiveness of Side Effects with Escitalopram and Duloxetine in Treatment-Naïve Depressed Adults

**DOI:** 10.3390/bs8070064

**Published:** 2018-07-17

**Authors:** Philip E. Polychroniou, Helen S. Mayberg, W. Edward Craighead, Jeffrey J. Rakofsky, Vivianne Aponte Rivera, Ebrahim Haroon, Boadie W. Dunlop

**Affiliations:** 1Emory University School of Medicine, Atlanta, GA 30322, USA; philip.polychroniou@emory.edu; 2Department of Psychiatry and Behavioral Sciences, Mount Sinai School of Medicine, New York, NY 10029, USA; helen.mayberg@mssm.edu; 3Department of Psychiatry and Behavioral Sciences, Emory University School of Medicine, Atlanta, GA 30329, USA; ecraigh@emory.edu (W.E.C.); jrakofs@emory.edu (J.J.R.); eharoon@emory.edu (E.H.); 4Department of Psychology, Emory University, Atlanta, GA 30329, USA; 5Department of Psychiatry and Behavioral Sciences, Tulane University School of Medicine, New Orleans, LA 70112, USA; aponte@tulane.edu

**Keywords:** adverse drug reaction, drug toxicity, antidepressant, serotonin uptake inhibitors, medication adherence

## Abstract

Side effect profiles of antidepressants are relevant to treatment selection and adherence among patients with major depressive disorder (MDD), but several clinically-relevant characteristics of side effects are poorly understood. We aimed to compare the side effect profiles of escitalopram and duloxetine, including frequencies, time to onset, duration, dose responsiveness, and impact on treatment outcomes. Side effects occurring in 211 treatment-naïve patients with MDD randomized to 12 weeks of treatment with flexibly-dosed escitalopram (10–20 mg/day) or duloxetine (30–60 mg/day) as part of the Predictors of Remission in Depression to Individual and Combined Treatments (PReDICT) study were evaluated. Escitalopram- and duloxetine-treated patients experienced a similar mean number of overall side effects and did not differ in terms of the specific side effects observed or their temporal profile. Experiencing any side effect during the first 2 weeks of treatment was associated with increased likelihood of trial completion (86.7% vs. 73.7%, *p* = 0.045). Duloxetine-treated patients who experienced dry mouth were significantly more likely to achieve remission than those who did not (73.7% vs. 44.8%, *p* = 0.026). Side effects that resolved prior to a dose increase were unlikely to recur after the increase, but only about 45% of intolerable side effects that required a dose reduction resolved within 30 days of the reduction. At the doses used in this study, escitalopram and duloxetine have similar side effect profiles. Understanding characteristics of side effects beyond simple frequency rates may help prescribers make more informed medication decisions and support conversations with patients to improve treatment adherence.

## 1. Introduction

Major depressive disorder (MDD) affects 15 million Americans over 18 years of age, and it is estimated that 13% of US adults are currently prescribed antidepressant medications [1]. Because marketed antidepressants have relatively similar efficacy [2], the side effect profiles of the individual medications frequently become a driving factor for medication selections by physicians and patients. Between 32–60% of patients discontinue their medications within the first 3 months of treatment, with side effects frequently cited as the main contributing factor [3,4,5].

Selective serotonin-receptor inhibitors (SSRIs) and serotonin-norepinephrine reuptake inhibitors (SNRIs), have emerged as the first-line medication options for treating depression, in large part because they have a more benign side effect profile and are safer in overdose than older tricyclic antidepressants (TCAs) and monoamine oxidase inhibitors [6]. Escitalopram is the S-enantiomer of racemic citalopram, with a high affinity for the serotonin transporter (SERT) and low affinity for other monoamine transporters or receptors [7]. Duloxetine is an SNRI believed to exert its clinical effect by modulating the serotonin (5-HT) and norepinephrine (NE) activity in the central nervous system. Duloxetine’s affinity for the SERT is roughly 10-fold greater than for the norepinephrine transporter, suggesting higher doses are required to get significant noradrenergic effect [8]. 

The package inserts of the trade name medications for escitalopram and duloxetine state that the most common treatment-emergent adverse event in patients with MDD was nausea, with a frequency of 15% and 23%, respectively. For escitalopram, other adverse events listed in descending order of frequency are: insomnia and ejaculation disorder (9%), diarrhea (8%), somnolence and dry mouth (6%), and dizziness, hyperhidrosis, fatigue, rhinitis, and influenza-like symptoms (5%) [9]. For duloxetine, the most common adverse events after nausea are headache and dry mouth (14%), constipation, diarrhea, fatigue, dizziness, somnolence, and insomnia (9%), decreased appetite and hyperhidrosis (6%), and abdominal pain (5%) [10]. However, because these data were collected across many trials conducted by different groups of investigators, comparisons of side effect frequencies between medications is uncertain.

A recent pooled analysis of trials comparing the tolerability of duloxetine versus other antidepressants found that duloxetine patients were more likely to drop out due to side effects than patients treated with escitalopram [11]. Among the individual adverse events, only nausea/vomiting occurred significantly more often among duloxetine-treated versus SSRI-treated patients; no adverse events were more common with SSRIs than duloxetine. These data suggest that the overall side effect profiles of duloxetine and SSRIs are similar.

Beyond the issue of differing frequencies of side effects between SSRIs and SNRIs, there remain many questions about antidepressant adverse event profiles that have high clinical relevance yet remain unexamined. These include: (1) the usual duration of antidepressant-related side effects; (2) the time to emergence of specific side effects after starting treatment; (3) the frequency of side effects emerging following drug initiation or dose increase; (4) the frequency with which antidepressant-induced side effects resolve after dose reduction; (5) the frequency with which side effects that have resolved will re-emerge after a dose increase; (6) the association between side effect occurrence with an antidepressant and probability of treatment outcomes (e.g., response, drop-out); (7) the effect that MDD severity or comorbid anxiety disorder has on reported side effects; and (8) the relationship between blood drug concentrations and side effect occurrence.

To address these questions, we analyzed the side effect data collected as part of the Predictors of Remission to Individual and Combined Treatments (PReDICT) study. The trial randomized 229 patients with MDD in a double-blind manner to the SSRI escitalopram or the SNRI duloxetine for 12 weeks of treatment.

## 2. Methods

### 2.1. Study Design

The design and rationale of the PReDICT study [12] and the primary clinical outcomes [13] have been previously published and are reviewed briefly here. PReDICT enrolled 344 adult men and women aged 18–65 who met DSM-IV criteria for a current non-psychotic major depressive episode [14] and who had never previously received an evidence-based treatment for MDD. Eligible patients had to score ≥18 at screening and ≥15 at baseline on the 17-item Hamilton Depression Rating Scale (HDRS) [15]. In addition, the Hamilton Anxiety Rating Scale (HAMA) [16] was performed at baseline and at each HDRS rating visit. Key psychiatric exclusion criteria included lifetime history of bipolar disorder, primary psychotic disorder, or dementia, or a current diagnosis of obsessive-compulsive disorder, eating disorder, dissociative disorder or substance abuse or dependence. Patients with significant uncontrolled medical conditions or any condition that could interfere with the study or the interpretation of the study results were excluded. Patients were also excluded if they had ever (lifetime) received treatment with any antidepressant medication for at least 4 weeks at a minimally effective dose. Additionally, any lifetime exposure (i.e., a single dose) to citalopram, escitalopram, or duloxetine was exclusionary. All patients provided written informed consent prior to beginning study procedures and the study was approved by the Emory Institutional Review Board and the Grady Hospital Research Oversight Committee. The study is registered on Clinicaltrials.gov, NCT00360399.

### 2.2. Treatments

Patients were randomized 1:1:1 to one of three possible treatments: (1) escitalopram 10–20 mg/d; (2) duloxetine 30–60 mg/d; or (3) cognitive behavior therapy. This report evaluates only those patients assigned to one of the two medication arms. The study medications were compounded in the Emory Investigational Drug Service and packaged into opaque purple capsules containing the equivalent of either 10 mg of escitalopram or 30 mg of duloxetine. Patients started with one capsule per day, which could be increased to two capsules once daily if the patient had not meaningfully improved by week 3. If the response to treatment plateaued during the trial, or if remission was not achieved by week 6, an increase to two capsules per day was required per the study protocol. Adverse events that were intolerable were addressed by lowering the dose back to one capsule per day. Dates for drug initiation and dose changes were recorded. Blood for pharmacokinetic quantification of antidepressant levels was collected at the week 2 visit, when both medications were expected to have achieved initial dose steady-state serum concentrations.

### 2.3. Assessment of Side Effects

Six physicians assessed patients for side effects during the study. All six were trained in side effect assessment by the lead study physician (B.W.D.). To be considered a side effect, the patient had to report an untoward physical or mental experience that either: (1) was a new event the patient had not previously experienced; or (2) reflected worsening of a pre-existing problem, either in terms of frequency or severity. Intensity was classified as “mild” (noticeable, but no more than minimally distressing, non-interfering with activities, and not requiring change in treatment), “moderate” (moderately distressing or causing some interference with activities, adjunctive treatment or dose adjustment may be required), or “severe” (significant interference in important activities, adjunctive treatment or dose reduction/drug cessation required). No formal assessment to measure inter-rater agreement on side effects between the physicians was conducted.

After randomization, patients returned at weeks 1–6, 8, 10, and 12 to complete the symptom rating scales and for a study physician to assess the degree of improvement and treatment tolerability. At each of these visits, patients were asked by a study physician if they had experienced any negative changes in their physical or mental health, or if they had experienced anything they might consider a side effect of treatment since the prior visit. Responses were recorded, documenting the start date, end date, severity, and the physician’s assessment of relatedness to the study medication (not related, unlikely, possibly, probably, or definitely). The relatedness of the study medication to the side effect was based on the physician’s evaluation of the potential precipitating factors, timing, concomitant medications, and pre-treatment status of the reported side effect. At each subsequent visit, the study physician also inquired about the status of each ongoing side effect, to determine whether it had abated or changed in severity. Side effect durations were calculated by subtracting the end date from the start date; for side effects that were not reported to have resolved, the patient’s last study visit date was used as the end date. Upon study completion, adverse events were coded by a physician (BWD) into version 18.0 of the Medical Dictionary for Regulatory Activities terms.

### 2.4. Statistical Methods

To minimize confounding arising from patient-reported health changes that had a low likelihood of being related to the treatment intervention, only side effects rated by the study physician as possibly, probably, or definitely related to the study medication were analyzed. Because few side effects were rated “severe,” we followed the approach used in prior studies and analyzed side effects only as present or absent, regardless of assessed severity [17]. Severe depression was defined by a baseline HDRS ≥ 20, with scores of 15–19 classified as non-severe [18]. *T*-tests were used to compare mean differences in normally distributed continuous data; times to onset and durations of side effects were found to be non-normally distributed and were compared using Mann-Whitney *U*-tests. Categorical outcomes of side effect proportions, drop-out, response and remission were compared using Chi-Square tests. Pearson’s correlation coefficient was used to examine the association between baseline HAMA scores with the number of reported side effects. Due to the exploratory nature of these analyses, corrections for multiple comparisons were not applied.

## 3. Results

### 3.1. Participants

Of the 229 patients randomized to medication, 18 never returned after randomization (9 in each medication group) and were not considered further in the analysis, leaving 211 total patients in the intent-to-treat sample (105 escitalopram, 106 duloxetine) and 178 completers (92 escitalopram, 86 duloxetine). Mean endpoint doses of the medications were 16.2 ± 5.1 mg/day for escitalopram and 48.0 ± 15.0 mg/d for duloxetine. Table 1 presents the baseline demographic and clinical characteristics of the analyzed patients.

### 3.2. Side Effect Frequencies and Durations by Treatment

In the intent-to-treat sample there were 793 instances of side effects considered to be possibly, probably, or definitely related to the study medication during the 12-week treatment period. Six-hundred-and-forty of these instances were classified into 20 specific side effects that occurred in ≥5% of the patients. Table 2 presents the frequencies and mean durations of these 20 side effects. If the same patient experienced multiple (recurrent) side effects of the same type, only the instance with the longest duration was used in the duration calculations. For the duration data, only the completer sample was used to avoid any bias toward shorter duration measures that would result from including early-terminating patients whose side effects had not resolved by the time of drop-out. Appendix A presents similar information for 11 additional side effects that are of potential clinical interest that occurred in <5% of patients.

The mean number of unique side effects did not differ between escitalopram- and duloxetine-treated patients, either in the intent-to-treat sample (3.17 ± 2.24 vs. 3.61 ± 2.38, respectively, *p* = 0.166) or among completers (3.28 ± 2.16 vs. 3.83 ± 2.38, respectively, *p* = 0.112). The variables that differed between treatment groups at baseline (HAMA score, age of onset of depression, and current age) did not significantly moderate these results. As shown in Table 2, none of the individual side effects significantly differed in frequency or duration between the treatments.

Among the 640 instances of the side effects listed in Table 2, 452 (70.6%) resolved within 30 days, which did not significantly differ between the medications (escitalopram: 202/296, 68%; duloxetine: 250/344, 73%; *p* = 0.220). Jitteriness was the shortest-lived side effect for both medications, with a mean duration of about 1 week. In contrast, several side effects persisted for a mean duration of >2 months, including sexual dysfunction, bruxism, and insomnia for escitalopram-treated patients, and sexual dysfunction, dry mouth, decreased appetite, increased sweating, and restlessness for duloxetine-treated patients.

### 3.3. Time to Onset of Side Effects

Table 3 presents the mean time to onset of the 20 most common side effects with the two medications. Among the 105 escitalopram patients, there were 301 separate side effects. Three side effects had an average time to onset within the first week of treatment: dyspepsia, dry mouth, and decreased appetite, and nine others had onset during the second week. Among the 106 duloxetine patients, there were 339 instances of the 20 most common side effects. Two side effects had a mean time to onset in the first week: jitteriness and dry mouth, and 11 others had onset during the second week. Time to onset did not differ for any of these side effects between the treatment groups (all *p* > 0.1). 

### 3.4. Impact of Dose Changes on Side Effects

To determine the effect of dose increases on side effect occurrence, we compared the counts of side effects beginning within 21 days of drug initiation versus those beginning within 21 days of a dose increase. In total, 573 side effects occurred within the first 21 days of drug initiation, (mean: 2.74/patient). In contrast, 118 side effects occurred within 21 days following a dose increase among the 155 patients who had their dose increased (mean: 0.76/patient). Side effects with drug initiation occurred significantly more frequently among duloxetine- than escitalopram-treated patients (Duloxetine: 318/413, 77.0%; Escitalopram: 255/372, 68.5%; *p* = 0.008). Conversely, following a dose increase, side effects were more common among escitalopram- than duloxetine-treated patients (Escitalopram: 66/292, 22.6%; Duloxetine: 52/319, 16.3%; *p* = 0.049).

To examine the recurrence of side effects after a dose increase, we identified those side effects that had onset after treatment initiation and subsequently resolved prior to a dose increase. There were 33 side effects in 27 individuals that recurred after a dose increase. The most frequently recurring side effects were headache (6/32, 18.6%) and nausea (6/38, 15.8%). Thus, once a side effect had resolved, the likelihood of the same side effect recurring after dose increase was low.

Nineteen subjects (10.7% of the completers) had their dose reduced due to side effects. Of the 33 total side effects present at the time of dose reduction, 18 (54.5%) of them resolved within 30 days following the dose change. The probability of resolution within 30 days of dose reduction was unrelated to the type of side effect and did not differ between the medications.

### 3.5. Impact of Side Effects on Treatment Outcomes

Patients who experienced a side effect within the first 2 weeks of beginning antidepressant treatment were significantly *more* likely to complete the full 12-week treatment than patients who did not (150/173, 86.7% versus 28/38, 73.7%, respectively, χ^2^ = 4.004, *p* = 0.045). Analyzing the two treatments separately found that this association was significant in the duloxetine group (*p* = 0.024), but not in the escitalopram group (*p* = 0.409). 

Despite being more likely to complete treatment, patients who completed treatment and who reported a side effect in the first 2 weeks were no more likely to achieve remission at week 12 than those who did not experience an early side effect, which was true for the whole sample combined (71/150, 47.3% vs. 11/28, 39.3%, respectively, χ^2^ = 0.615, *p* = 0.433), as well as for each treatment individually. Patients experiencing dry mouth during the first 2 weeks were more likely to achieve remission (21/33, 63.6%) compared to those who did not (61/145, 42.1% χ^2^ = 5.033, *p* = 0.025). In contrast, restlessness was negatively associated with remission (0/5, 0.0% vs. 82/173, 47.4%, respectively, χ^2^ = 4.394, *p* = 0.036). Within-treatment analyses identified no side effects in the escitalopram group that predicted remission. However, among duloxetine-treated patients, dry mouth was predictive of remission (14/19, 73.7% vs. 30/67, 44.8%, respectively, χ^2^ = 4.951, *p* = 0.026), and headache predicted non-remission (4/17, 23.5% vs. 40/69, 58.0%, respectively, χ^2^ = 6.475, *p* = 0.011). 

### 3.6. Impact of Depression Severity and Anxiety on Reported Side Effects

The sample was evenly divided between severe (*n* = 105) and non-severe (*n* = 106) depression based on the baseline HDRS score. The mean number of side effects experienced during the study did not differ between the severe and non-severely ill patients (3.4 ± 2.1 and 3.4 ± 2.5, respectively; *p* = 0.986). 

Anxiety as assessed by baseline HAMA scores was also not significantly associated with the number of side effects (*r* = 0.029, *p* = 0.68). Similarly, the mean number of side effects did not significantly differ between patients who did (*n* = 88) or did not (*n* = 123) have a current comorbid anxiety disorder (3.7 ± 2.7 versus 3.2 ± 1.9, *p* = 0.087).

### 3.7. Relationship between Plasma Drug Concentration and Side Effects

To examine whether the emergence of side effects early in treatment was associated with the level of drug exposure, we examined correlations between the number of side effects reported at week 2 and the serum concentrations of the antidepressants at week 2. The correlations were not significant for either medication (Figure 1). We also compared the drug serum concentrations at week 2 among patients who did or did not experience one of the six most common side effects (nausea, headache, somnolence, sexual dysfunction, fatigue, and insomnia) that occurred within the first 2 weeks of treatment. As shown in Figure 2, the duloxetine concentration was significantly higher in patients who experienced nausea versus those who did not (42.1 ± 54.9 ng/mL versus 20.6 ± 17.3 ng/mL, respectively, *p* = 0.023); all other comparisons were not significant.

## 4. Discussion

This analysis of 211 treatment-naïve depressed adults randomly assigned to 12 weeks of treatment with escitalopram or duloxetine found similar side effect profiles for the two medications. These results indicate that in the dose range employed in the study, duloxetine was acting primarily as an SSRI. Side effects most consistently emerging in the first week of treatment included jitteriness and the gastrointestinal effects of dyspepsia (abdominal pain), decreased appetite, and dry mouth. Experiencing a side effect during the first 2 weeks of treatment was positively associated with completing the 12-week trial. Restlessness, though uncommon, predicted non-remission to treatment, and dry mouth predicted remission, particularly for duloxetine-treated patients. Re-emergence of specific side effects after a dose increase was uncommon if a side effect had fully resolved prior to the increase in dose. Taken together, these results provide additional information for prescribers to use in discussions with patients about the probability, duration, and dosing approaches regarding side effects occurring with escitalopram and duloxetine. 

An unexpected finding was that occurrence of side effects in the first 2 weeks predicted completion of the full 12-week trial; this overall effect was driven by a significant difference in the duloxetine arm, which was absent in the escitalopram arm. One interpretation of this finding is that the experience of side effects may signal to the patient that they are on a “powerful” medicine that will affect them, which sustains willingness to adhere to treatment, especially if side effects are not severe. In PReDICT, the active inquiry about side effects made by study physicians at each visit may have enhanced the sense of treatment collaboration with the patients, thereby supporting trial completion. These data are consistent with prior studies indicating that physician engagement with patients about side effects improves treatment adherence [19,20].

Although experiencing any side effect in the first 2 weeks did not predict remission, the early emergence of dry mouth was significantly associated with remission, and restlessness was predictive of non-remission. One prior study specifically associated the side effect of dry mouth with patients who responded to antidepressants between weeks 4–6 of treatment compared to those who responded by week 2 [21]. Given that salivation is regulated by cholinergic and noradrenergic tone, it is possible that the association of dry mouth with remission may reflect an individual’s sensitivity to the modulating effect of an antidepressant on these monoamine systems.

Duloxetine-treated patients who experienced nausea had higher plasma concentrations of the medication than those who did not experience this side effect. A meta-analysis of placebo-controlled randomized trials found nausea to be the most common symptom reported among duloxetine-treated patients across all indications [22]. Similarly, a pooled analysis of head-to-head trials comparing duloxetine versus escitalopram for depression found only nausea was significantly more common among duloxetine-treated patients [11]. In the trials included in this analysis, duloxetine was always prescribed at a fixed dose of 60 mg/day from the first day of treatment [23,24,25]. In contrast, in PReDICT, patients were started at 30 mg/day, and the frequency of nausea did not differ from that observed in the escitalopram-treated patients. Although the package insert for the trade name version of duloxetine states, “For some patients, it may be desirable to start at 30 mg once daily for 1 week, to allow patients to adjust to the medication before increasing to 60 mg once daily,” this information has unclear clinical application because the likelihood of side effects cannot be known beforehand. The consistent associations across trials of nausea rates with higher initial duloxetine doses, along with the higher serum concentrations among patients experiencing nausea in the current analysis, suggest it would be prudent to start at <60 mg/day in all patients initiating duloxetine treatment.

Strengths of this analysis include that all patients in the study had never previously received an adequate duration of antidepressant medication, leading the study to be free of selection bias against one of the medications that could have arisen from patients’ prior treatment experiences. In addition, the detailed data around timing of onset, offset, and dose responsiveness of reported side effects provided a richness of side effect description mostly lacking from prior investigations into antidepressant tolerability.

A potential limitation of this analysis was the absence of a structured side effect rating questionnaire. Self-report questionnaires have the benefit of consistent administration and have been shown to capture a wider array and greater number of side effects than spontaneous reporting or physician inquiry [26,27]. However, the questionnaire approach to side effects suffers from several important limitations, including exclusion of relevant side effects (e.g., yawning, fatigue, emotional blunting), and inability to assess the relatedness of the side effects specifically to the medication. Furthermore, side effect questionnaires often capture aspects of psychiatric illness or unrelated health problems [26], introducing substantial noise into side effect assessment. Indeed, nearly all studies using self-report questionnaires have found that side effect frequency correlates with the severity of depression or anxiety [17,28,29,30]. One large study that employed a side effect checklist found that adverse reactions were more common when patients were medication free at baseline than during their antidepressant treatment [17]. The conflicting results regarding the association of side effects with the level of depression severity and anxiety between our analysis and those of investigators using self-report questionnaires suggests that associations between psychiatric symptom severity and side effect occurrence is primarily due to increased distress expression [31] or heightened sensitivity and attention to uncomfortable physical sensations associated with depression [32,33,34]. Therefore, by evaluating only those side effects considered by the study physicians as being possibly resulting from the medication, the current analysis reduced the potential for confounding arising from self-report questionnaire data. A second potential limitation was that the mean dose of duloxetine used in the study, 48 mg/day, was lower than doses typically used by psychiatrists to treat MDD, though it is within the 40–60 mg/day dose indicated on the medication label [10]. Duloxetine was also started at a slightly sub-threshold dose (30 mg/day) compared to escitalopram (10 mg/day). A higher starting dose or mean dose of duloxetine may have resulted in more frequent noradrenergic-related side effects. Although patients were evaluated for abuse of alcohol or use of illicit substances during the screening phase of the study, they were not re-evaluated for these factors after they initiated treatment.

## 5. Conclusions

Although the use of physician-assessed side effects runs the risk of incomplete capture of all side effects, there are substantial advantages to this approach in reducing false positives and allowing greater signal-to-noise ratio in the collected data. Starting duloxetine at 30 mg/day is a prudent approach to minimize nausea for many patients. Side effects occurring during treatment need not lead to high rates of treatment drop-out if prescribers dose medications judiciously and regularly attend to patients’ experiences.

## Figures and Tables

**Figure 1 behavsci-08-00064-f001:**
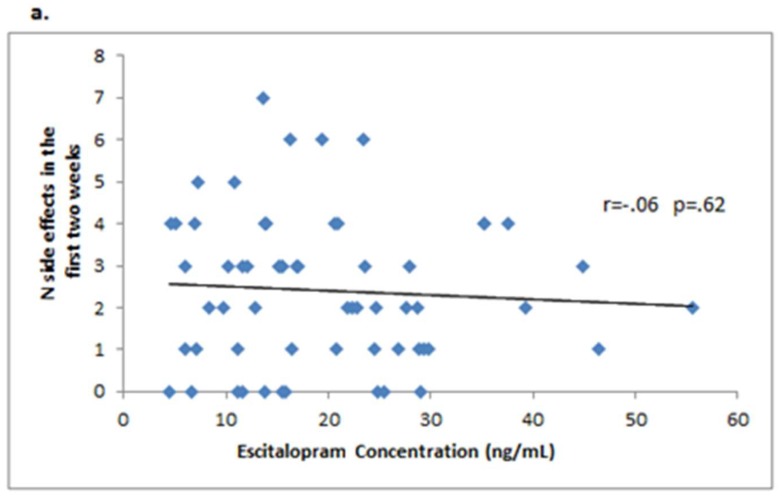
Lack of correlation between serum plasma concentrations of escitalopram or duloxetine and the number of side effects reported at week 2 of treatment. (**a**) Escitalopram; (**b**) Duloxetine.

**Figure 2 behavsci-08-00064-f002:**
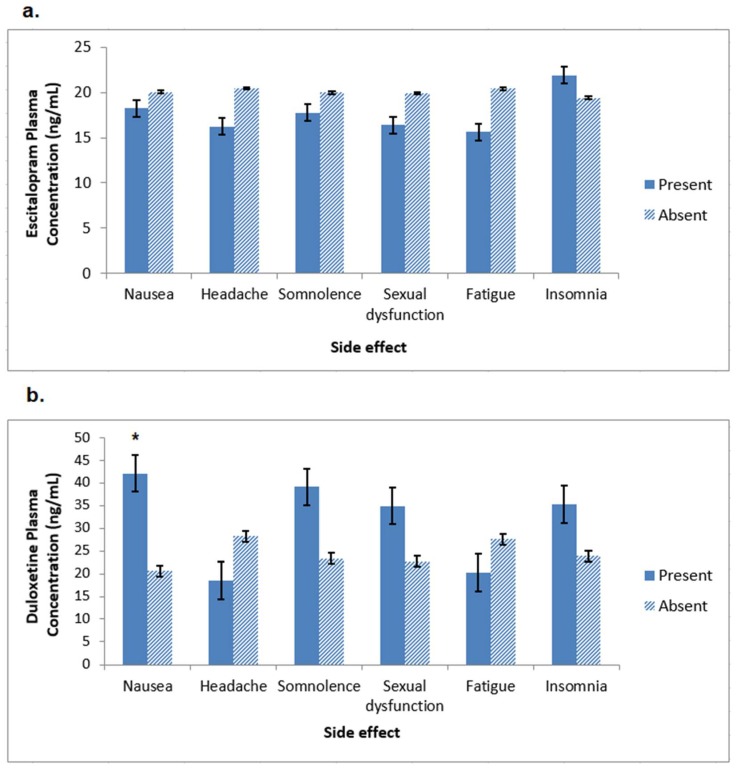
Mean serum antidepressant concentrations at week 2 for the six most common side effects occurring within the first 2 weeks of treatment. (**a**) Escitalopram; (**b**) Duloxetine * *p* = 0.023.

**Table 1 behavsci-08-00064-t001:** Clinical and demographic characteristics at baseline.

Characteristic	Escitalopram (*n* = 105)		Duloxetine (*n* = 106)		*F*	*p*
	Mean	SD	Mean	SD		
Age (yrs)	41.1	12.1	38.2	11.4	3.904	**0.049**
Age at first episode (yrs)	32.8	15.1	27.7	13.4	6.850	**0.010**
Current episode duration (wks)	103.8	159.1	123.9	237.7	0.514	0.474
HDRS	20.0	3.6	19.4	3.8	1.766	0.185
BDI	23.3	7.2	23.3	7.3	0.000	0.998
HAMA	16.6	5.0	15.1	5.1	4.335	**0.039**
	*N*	%	*N*	%	*X* ^2^	*P*
Sex					0.379	0.538
Male	49	46.7	45	42.5		
Female	56	53.3	61	57.5		
Race					1.892	0.864
White	46	43.8	53	50.0		
Black	22	21.0	20	18.9		
Other or Multiple	37	35.2	33	31.1		
Ethnicity					0.119	0.730
Hispanic	33	31.4	31	29.2		
Non-Hispanic	72	68.6	75	70.8		
Current Anxiety Disorder					0.114	0.736
Yes	45	42.9	43	40.6		
No	60	57.1	63	59.4		
Previous Episodes					2.717	0.257
1	57	54.8	47	44.3		
2	17	16.3	25	23.6		
≥3	30	28.8	34	32.1		
Chronic Episode (≥2 yrs)	29	28.2	34	32.4	0.440	0.507

Bolded values represent *p* < 0.05. BDI: Beck Depression Inventory; HAMA: Hamilton Anxiety Rating Scale HDRS; Hamilton Depression Rating Scale.

**Table 2 behavsci-08-00064-t002:** Frequency and duration of side effects among patients treated with escitalopram or duloxetine.

Adverse Event	Frequency							Duration				
	Total		Escitalopram		Duloxetine		*p*-Value	Escitalopram		Duloxetine		*p*-Value
	*(n* = 211)		(*n* = 105)		(*n* = 106)			(*n* = 92)		(*n* = 96)		
	*n*	%	*n*	%	*n*	%		Mean (Days)	SD	Mean (days)	SD	
Nausea	58	27.5	26	24.8	32	30.2	0.38	7.7	8.5	17.6	36.0	0.20
Headache	53	25.1	27	25.7	26	24.5	0.84	20.8	22.7	12.7	17.4	0.19
Somnolence	48	22.7	23	21.9	25	23.6	0.77	44.0	63.3	19.2	17.2	0.08
Sexual dysfunction	47	22.3	23	21.9	24	22.6	0.90	89.0	93.7	88.5	70.0	0.98
Fatigue	46	21.8	22	21	24	22.6	0.77	49.1	55.2	31.4	32.5	0.22
Insomnia	45	21.3	20	19	25	23.6	0.42	71.5	116.4	37.2	37.3	0.20
Dry mouth	41	19.4	16	15.2	25	23.6	0.13	54.4	62.9	72.3	73.4	0.43
Dizziness	33	15.6	16	15.2	17	16	0.87	38.9	78.4	10.3	11.4	0.19
Diarrhea	32	15.2	15	14.3	17	16	0.72	21.0	34.6	13.4	22.3	0.48
Constipation	20	9.5	8	7.6	12	11.3	0.36	24.6	30.0	27.2	22.6	0.83
Anxiety	18	8.5	8	7.6	10	9.4	0.64	15.8	18.3	15.4	16.9	0.97
Abdominal pain	17	8.1	6	5.7	11	10.4	0.21	12.6	15.1	9.6	9.6	0.64
Yawning	16	7.6	7	6.7	9	8.5	0.62	31.6	16.5	45.0	59.9	0.58
Appetite decreased	16	7.6	7	6.7	9	8.5	0.62	20.2	15.6	50.0	53.2	0.22
Emotional blunting	15	7.1	11	10.5	4	3.8	0.06	38.0	24.4	24.8	24.8	0.38
Dyspepsia	14	6.6	6	5.7	8	7.5	0.59	33.8	42.1	13.8	11.8	0.34
Bruxism	14	6.6	8	7.6	6	5.7	0.57	88.6	85.6	54.0	41.6	0.38
Sweating increased	13	6.2	4	3.8	9	8.5	0.16	41.3	23.4	41.3	37.1	0.99
Jitteriness	12	5.7	6	5.7	6	5.7	0.99	10.3	5.1	5.8	5.9	0.28
Restlessness	11	5.2	5	4.8	6	5.7	0.77	18.3	24.5	19.3	14.7	0.95

**Table 3 behavsci-08-00064-t003:** Time to onset of side effects (Days).

Side Effect	All Patients		Escitalopram		Duloxetine		*p*-Value
	(*n* = 793)		(*n* = 372)		(*n* = 421)		
	Mean	SD	Mean	SD	Mean	SD	
Dry mouth	5.1	7.8	5.0	6.4	5.2	8.6	0.93
Dyspepsia	6.6	10.6	5.0	7.8	7.9	12.8	0.64
Appetite decreased	6.7	14.2	5.7	9.7	7.3	17.2	0.83
Jitteriness	8.6	13.3	12.0	15.9	5.1	10.1	0.36
Yawning	9.8	13.7	10.6	13.5	9.1	14.5	0.82
Nausea	10.1	17.6	10.1	14.7	10.1	20.0	0.99
Restlessness	10.8	14.3	10.8	14.6	10.7	15.4	0.99
Diarrhea	12.1	18.3	13.3	15.4	11.2	20.7	0.75
Fatigue	12.6	18.2	13.2	17.6	12.1	19.1	0.83
Anxiety	12.6	19.3	16.4	21.3	9.2	17.7	0.43
Abdominal pain	13.3	12.8	13.5	13.3	13.2	13.1	0.96
Headache	13.8	18.2	15.2	18.0	12.1	18.5	0.49
Constipation	14.3	14.7	17.3	17.7	12.3	12.6	0.43
Insomnia	14.8	16.8	15.3	16.3	14.4	17.6	0.86
Dizziness	15.6	19.5	12.6	15.2	18.6	23.1	0.35
Somnolence	15.6	19.2	19.2	20.4	12.2	17.7	0.19
Sweating increased	19.0	15.9	13.3	12.3	21.6	17.3	0.41
Bruxism	19.9	19.8	20.2	19.7	19.6	21.5	0.95
Sexual dysfunction	20.9	21.1	25.5	21.5	16.7	20.1	0.11
Emotional blunting	23.5	18.0	20.9	15.2	30.5	25.6	0.38

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
