# Peer review of "Temporal Profiles and Dose-Responsiveness of Side Effects with Escitalopram and Duloxetine in Treatment-Naïve Depressed Adults"

_behavsci, 2018, doi:10.3390/bs8070064_

Round 1
Reviewer 1 Report
Comments and suggestions for authors
Polychroniou and colleagues have conducted an interesting research about the side effects of escitalopram and duloxetine, two of the most used antidepressants, in depressed adults who have not received treatment yet. The main strength of this study resides in this valuable sample precisely, not only highly difficult to obtain but also seemingly relevant to depression research. Moreover, the authors, recognizing their study limitations and by means of well-explained rational procedures, reach significant conclusions like the 30 mg/kg starting duloxetine dose to minimize nausea. Despite the lack of a large amount of impressive results, the manuscript is clear and descriptive. However, some comments regarding the study design and variable control should be taken into consideration.
· The exclusion criteria did not mention comorbidities with inflammatory diseases. Given the vast array of evidences linking depressive disorders, antidepressant response and inflammation, the authors should provide rationale for not excluding these patients or, at least, controlling the possible influence of this variable.
· Lifetime treatment with citalopram, escitalopram or duloxetine was exclusionary, but there is no mention of other monoaminergic drugs which could represent a determinant bias in the study. It is stated that the subjects had never previously received an evidence-based treatment for MDD, but they could receive these treatments for other reasons. Please, clarify this information.
· No method, e.g. blood determinations, was described as a follow-up of substances that could interfere with the treatment (other drugs or abuse substances like alcohol). Is there any assessment of their absence during the treatments? If not, this represents another limitation and should be considered in the discussion section.
· In the clinical and demographic characteristics of patients, significant differences in age, age at first episode and HAMA score were found between escitalopram and duloxetine groups. A statistical analysis controlling for these variables is relevant to evaluate their influence in the results.
Minor issues
· Figures 2a and 2b do not include error bars on the graph. Please, draw them to enhance accuracy and make it more representative and intuitive.
Author Response
Response to Reviewers
We thank the reviewers for their careful reading of the manuscript, and the editor for offering us the chance to respond to these comments. Wherever possible we have modified the manuscript to incorporate the reviewers’ suggestions. Our point-by-point response to the reviewers is below.
Reviewer 1
1. The exclusion criteria did not mention comorbidities with inflammatory diseases. Given the vast array of evidences linking depressive disorders, antidepressant response and inflammation, the authors should provide rationale for not excluding these patients or, at least, controlling the possible influence of this variable.
Response:
For purposes of space, we did not list the full exclusion criteria for the PReDICT study in this manuscript because they have been previously published and are available in an open access journal (Dunlop et al., Trials, 2012). Any patient with an uncontrolled medical condition, or any controlled condition that could confound interpretation of the study results, were excluded, including patients with inflammatory diseases (e.g. lupus erythematosus), congestive heart failure, cerebrovascular accidents, active hepatitis, etc. To address the reviewer’s concern about inflammatory diseases, as well as those other readers may have about other types of medical disorders, we have added to the Methods the following sentence:
“Patients with significant uncontrolled medical conditions, or any condition that could interfere with the study or the interpretation of the study results.”
2. Lifetime treatment with citalopram, escitalopram or duloxetine was exclusionary, but there is no mention of other monoaminergic drugs which could represent a determinant bias in the study. It is stated that the subjects had never previously received an evidence-based treatment for MDD, but they could receive these treatments for other reasons. Please, clarify this information.
Response:
An adequate course of treatment for depressed mood with any antidepressant (monoaminergic or other mechanism) was exclusionary; our explicit mention of citalopram, escitalopram and duloxetine was made because ANY treatment (even a single dose) with any of these three medications was additional exclusion criteria. We realize that our description of this aspect of the study in section 2.1 was inadequately clear. We have rewritten this section to clarify the above information as follows:
“Patients were excluded if they had ever (lifetime) received treatment for depressed mood with any antidepressant medication for at least four weeks at a minimally effective dose. Additionally, any lifetime exposure (i.e., a single dose) to citalopram, escitalopram, or duloxetine was exclusionary”
Only 25 of the 344 patients had ever received even a single dose of an antidepressant medication, including two who took trazodone for insomnia for less than 2 weeks. Only four patients had ever taken an antidepressant medication for >30 days, for the indications of panic disorder, premenstrual tension, anxiety, and insomnia.
3. No method, e.g. blood determinations, was described as a follow-up of substances that could interfere with the treatment (other drugs or abuse substances like alcohol). Is there any assessment of their absence during the treatments? If not, this represents another limitation and should be considered in the discussion section.
Response:
Although we did collect blood tests to assure absence of illicit drug use or excessive alcohol use at the screening visit, we did not collect these tests at later points in the trial. We have added the following sentence to the Discussion:
“Although patients were evaluated for abuse of alcohol or use of illicit substances during the screening phase of the study, they were not reevaluated for these factors after they initiated treatment.”
4. In the clinical and demographic characteristics of patients, significant differences in age, age at first episode and HAMA score were found between escitalopram and duloxetine groups. A statistical analysis controlling for these variables is relevant to evaluate their influence in the results.
Response:
The three variables (HAMA score, age, and age of first episode) had very weak (all r < 0.11) and non-significant correlations with the number of side effects. Regression analyses also found that none of these variables significantly moderated the associations between treatment arm and number of side effects. In the prior version of the manuscript, we reported this lack of association for the HAMA and comorbid anxiety disorder variables. In the revised version, we have added the following sentence to section 3.2 of the Results, where we report the number of side effects per treatment arm:
“The variables that differed between treatment groups at baseline (HAMA score, age of onset of depression, and current age) did not significantly moderate these results.”
5. Figures 2a and 2b do not include error bars on the graph. Please, draw them to enhance accuracy and make it more representative and intuitive.
Response:
Error bars have been added to the figure.
Reviewer 2 Report
Dear Authors,
the presented results are interesting and give a broader idea to the problem of side effects occurring in patients treated with antidepressants.
I have some minor comments to your work:
- how about the medicine doctors examining the patients? how large was the group of doctors? how did the authors introduce unity of judgement? have the doctors obtained any information on how to classify the intensity of side effects in patients? please coment on that in the manuscript.
- how can the authors explain, that ca 50 patients left the trial? what could be the reason for this?
- line 287 and other places in the manuscript: please do not refer to the total number of patients but the actually studied group
Author Response
Response to Reviewers
We thank the reviewers for their careful reading of the manuscript, and the editor for offering us the chance to respond to these comments. Wherever possible we have modified the manuscript to incorporate the reviewers’ suggestions. Our point-by-point response to the reviewers is below.
Reviewer 2
1. How about the medicine doctors examining the patients? how large was the group of doctors? how did the authors introduce unity of judgement? have the doctors obtained any information on how to classify the intensity of side effects in patients? please coment on that in the manuscript.
Response:
The following paragraph has been added to the Methods in section 2.3.
“Six physicians assessed patients for side effects during the study. All six were trained in assessment of side-effects by the lead study physician (B.W.D.). To be considered a side effect, the patient’s reported experience had to be either: 1) a new event the patient had not previously experienced, or 2) reflect worsening of a pre-existing problem, either in terms of frequency or severity. Intensity was classified as “mild” (noticeable, but no more than minimally distressing, non-interfering with activities, and not requiring change in treatment), “moderate” (moderately distressing or causing some interference with activities, adjunctive treatment or dose adjustment may be required), or “severe” (significant interference in important activities, adjunctive treatment or dose reduction/drug cessation required). No formal assessment to measure inter-rater agreement on side effects between the physicians was conducted.’
2. How can the authors explain, that ca 50 patients left the trial? what could be the reason for this?
Response:
Of the 112 patients who started treatment with escitalopram, 26 terminated early. The reasons for early termination were adverse event (8), protocol violation (6), withdrew consent (6), lost to follow-up (5), and moved away (1). Of the 113 patients who started treatment with duloxetine, 34 terminated early: protocol violation (12), withdrew consent (8), lost to follow-up (7), adverse event (6), and moved away (1).
These data are already reported in the primary paper (Dunlop et al., 2017) and have not been added again here. The overall drop-out rate is similar to the rates observed in other 12 week studies of antidepressants.
3. Line 287 and other places in the manuscript: please do not refer to the total number of patients but the actually studied group
Response:
We believe the reviewer is referring here to the first line of the Discussion, where we state 229 were analyzed. We apologize for this oversight, and have changed the stated number to 211, which was the number of subjects available for the side effects analysis.
To clarify: The total N randomized in the PReDICT was 344, of which 229 were randomized to one of the two medications. The number of patients analyzed for side effects was 211, which was the number of patients who took at least one dose of study medication and returned for at least one follow-up visit. We have revised the manuscript to make this clearer throughout.
Round 2
Reviewer 1 Report
The authors have clarified all the concerns and the manuscript is now clearer and significantly improved.